# Molecular and Cellular Mechanisms of Cytotoxic Activity of Vanadium Compounds against Cancer Cells

**DOI:** 10.3390/molecules25071757

**Published:** 2020-04-10

**Authors:** Szymon Kowalski, Dariusz Wyrzykowski, Iwona Inkielewicz-Stępniak

**Affiliations:** 1Department of Medical Chemistry, Medical University of Gdansk, 80-211 Gdansk, Poland; szymon.kowalski@gumed.edu.pl; 2Faculty of Chemistry, University of Gdansk, 80-308 Gdansk, Poland; dariusz.wyrzykowski@ug.edu.pl

**Keywords:** vanadium compounds, cytotoxicity, cancer cells, molecular mechanisms, cellular mechanisms

## Abstract

Discovering that metals are essential for the structure and function of biomolecules has given a completely new perspective on the role of metal ions in living organisms. Nowadays, the design and synthesis of new metal-based compounds, as well as metal ion binding components, for the treatment of human diseases is one of the main aims of bioinorganic chemistry. One of the areas in vanadium-based compound research is their potential anticancer activity. In this review, we summarize recent molecular and cellular mechanisms in the cytotoxic activity of many different synthetic vanadium complexes as well as inorganic salts. Such mechanisms shall include DNA binding, oxidative stress, cell cycle regulation and programed cell death. We focus mainly on cellular studies involving many type of cancer cell lines trying to highlight some new significant advances.

## 1. Introduction

The discovery that metals are essential for the structure and function of biomolecules has given a completely new perspective on the role of metals in living organisms [1]. It has been determined that they can perform numerous processes that cannot otherwise be achieved. For instance, iron is essential for ribonucleotide reductase activity, an enzyme required for the rate limiting step of DNA synthesis [2]. Furthermore, over 300 enzymes that play important roles in gene expression include zinc in their structure (e.g., zinc-finger transcription factor) [3].

In the year 1965, Barnett Rosenberg serendipitously discovered the Pt(II) coordination compound, *cis*-[Pt(NH_3_)_2_Cl_2_] (cisplatin) [4], one of the most successful metal-based drugs. This happened during studies on the effect of electric currents on bacteria. It has been found that cell division was inhibited by the production of cisplatin from the platinum electrodes [4]. Further studies on this platinum(II) agent indicated that it possessed antitumor activity and cisplatin was approved by the FDA in 1978 for the treatment of ovarian and testicular cancer [5]. Moreover, two derivatives of cisplatin were approved for treatment: carboplatin in 1989 for ovarian cancer [6] and oxaliplatin in 2002 for advanced colorectal cancer [7]. Both compounds exhibit fewer side effects and therefore have a lower toxicity as well as better retention in the body relative to cisplatin [8,9]. Unfortunately, despite these benefits, platinum-based chemotherapy is accompanied by side effects such as vomiting, neuropathy or nephrotoxicity [10,11]. However, an upwards trend for the market for platinum-based anticancer drugs has been maintained [12].

Nowadays, the design and synthesis of new metal-based compounds, as well as metal ion binding components, for the treatment of human diseases is one of the main aims of bioinorganic chemistry [13]. Metal-based molecules exhibit a wide range of unique properties, which cannot be achieved by typical organic compounds, such as a large amount of coordination numbers, accessible redox states or kinetic and thermodynamic characteristics [13]. Examples include metals for imaging, such as a gadolinium complex for MRI contrast [14] or positron emitting metal for positron emission tomography (PET) [15]. Moreover, metal ions coordinated to the organic ligand change the flexibility as well as geometry of the resulting complexes, causing more effective exploration of the activity space of the molecular target. Such a situation was observed in the case of the interactions of octahedral pyridocarbazole ruthenium(II) or iridium(III) complexes with the ATP-binding site of a protein kinase [16].

This new approach to the design and synthesis of new metal-based molecules has not excluded vanadium, which is the 18th most abundant element in our planet’s crust and the 2nd most common element in sea water, in regard to transition metal concentration (between 30 and 35 nM), where it exists mainly in the form of H_2_VO_4_^−^ [17]. It is noteworthy that vanadium is also present in many living organisms including amanita mushrooms, marine Polychaeta fan worms or ascidians [17]. Importantly, vanadium deficiency in an animal diet produces many side effects: reduced fertility, increased rates of spontaneous abortion, decreased milk production and skeletal abnormalities [18]. Vanadium is constantly present in the human body in quantities of about 100 μg; however, it is not considered to be a micronutrient [17]. In the last 15 years, significant progress in the chemistry of vanadium has been made, particularly with regard to its therapeutic applications [19].

One of the areas of vanadium research is its potential anticancer activity. Recently, reviews describing its mechanism of anticancer activity have been published [19,20,21,22]. This current review aims to summarize more recent molecular and cellular mechanisms in the cytotoxic activity of many different synthetic vanadium complexes as well as inorganic salts. We focus mainly on cellular studies involving many types of cancer cell lines in an attempt to highlight some new significant advances.

## 2. Mechanisms of Cytotoxicity

### 2.1. DNA: The Classical Target

In classical chemotherapy, anticancer compounds directly target DNA, causing lesions and ultimately triggering cell death. This is in accordance with the cisplatin paradigm in which one of the major therapeutic pathways of the platinum-based complex is based on interaction with DNA to generate inter- and intra-strand crosslinks. This leads to transcription inhibition, disruption of the DNA repair system and ultimately to apoptosis [23]. Nowadays, it has been established that DNA is one of the primary pharmacological targets of many metal-based complexes [24]. The binding affinities of DNA-metal complexes are a key issue for understanding the mechanism of effective metal-based chemotherapeutic drugs.

Furthermore, in the case of vanadium, many studies on its interaction with DNA have been performed. Mohamadi et al. [25] have used electronic absorption spectroscopy, competitive fluorescence assay and cyclic voltametry studies to determine DNA binding activities. The obtained results showed groove binding of the mononuclear diketone-based oxido-vanadium(IV) complex (**1**) to the salmon sperm DNA, accompanied with a partial insertion of the ligand between the base stacks of the DNA. These experimental results have been confirmed by the results of molecular docking [25]. Additionally, the synthesized complex (**1**) exhibited cytotoxicity against breast, liver and colon cancer cell lines [25]. Another study on the diketone-based oxovanadium complexes (**2** and **3**) (containing trifluoropentanedione and trifluoro-1-phenylbutanedione) has shown that investigated complexes preferred minor groove binding with DNA [26]. Interestingly, a non-oxido vanadium(IV) complex with a catechol-modified 3,3′-diindolylmethane (**4**) exhibited stronger DNA binding than cisplatin [27]. Importantly, Fik et al. [28] have demonstrated that vanadium complexes with dimethylterpyridine (**5** and **6**) exhibited cytotoxic activity against human cervical carcinoma cells by direct interactions with DNA, thus increasing the level of arrest cells in stage G2/M. The DNA interaction ability has been determined also for the phenantroline vanadium complex (**7**–**10**) with simultaneous cytotoxic activity against human ovarian and breast carcinoma cells [29]. Furthermore, Rui et al. [30] has shown that vanadium complexes derived from thiosemicarbazones and fluoro-phenanthroline derivatives (**11**–**13**) interacted with calf-thymus DNA (CT-DNA) through a non-classical intercalative mode and they could efficiently cleavage plasmid pBR322 DNA upon exposure to ultraviolet light. Additionally, all investigated complexes exhibited anti-proliferative activity against many human tumor cell lines [30]. A similar study has been performed for oxidovanadium(IV) phenanthroimidazole derivatives (**14**–**17**), which could bind with CT-DNA and which cleaved supercoiled plasmid DNA in the presence of H_2_O_2_, and also exhibited cytotoxicity against a cervical cancer cell line by inducing apoptosis [31]. The DNA binding activity has been determined for many other synthetic complexes including the vanadium(V)-pyridylbenzimidazole complex (**18**) [32], mixed-ligand oxidovanadium(V) hydrazone complexes (**19** and **20**) [33] or VO(II)-Perimidine [1H-Benzo(de)quinazoline] (**21**–**25**) complexes [34].

An interesting approach to anticancer therapy provides photodynamic therapy (PDT) which is based on selectively damaging the photo-exposed cancer cells, leaving the unexposed healthy cells unaffected [35]. Kumar et al. [36] have designed an oxidovanadium(IV) complex with a 4,4-difluoro-4-bora-3a,4a-diaza-s-indacene (BODIPY)-based photosensitizer (**26** and **27**) for its PDT action, which showed dual activity: light-activated VO^2+^-DNA crosslink formation and singlet oxygen (^1^O_2_) induced mitochondria-targeted PDT. Interestingly, the BODIPY-based vanadium complex (**27**) exhibited remarkable photocytotoxicity against cervical and breast cancer cell lines via apoptotic pathway in visible light (400–700 nm) compared with low dark toxicity [36]. In other research, DNA melting and comet assay studies suggested the formation of DNA crosslinks by terpyridyl oxidovanadium(IV) complexes (**28** and **29**), and this effect was observed upon irradiation with visible light [37]. Additionally, neutral oxidovanadium(V) complexes with different organic ligands (**30**–**33**) had DNA binding propensity and it was shown that these interacted with CT-DNA through minor groove binding mode; however, the complex with isonicotinoylhydrazone of 2-hydroxy acetophenone (**32**) showed the highest photo-induced DNA cleavage activity [38].

Importantly, many indirect mechanisms that affect DNA structure and stability have been determined. Topoisomerases are enzymes that control the topological state of DNA through the re-joining or breaking of DNA strands [39]. There are two classes of topoisomerases: type I enzymes, which are able to transiently nick one of the two DNA strands, and type II enzymes which act by nicking both DNA strands and whose activity is ATP-dependent [39]. Research has shown that a oxidovanadium(IV) complex with silibinin (**34**) inhibited relaxation activity of human topoisomerase IB in a dose-dependent manner. However, the inhibition was incomplete, suggesting that the inhibitory effect of the vanadium compound is reversible [40].

The structure and activity of DNA-binding vanadium compounds are summarized in Table 1. Vanadium complexes may also cause indirect DNA damage by generating reactive oxygen species (ROS) resulting in oxidative stress. This is discussed in the following subsection.

### 2.2. Oxidative Stress

Oxidative stress is a complex issue [41]. This concept was first formulated in 1985 as “a disturbance in the prooxidant-antioxidant balance in favour of the former” [42]. The current definition takes into account the role of redox signaling and reads as follows: “An imbalance between oxidants and antioxidants in favour of the oxidants, leading to a disruption of redox signaling and control and/or molecular damage” [43]. The oxidants, which include free radicals, are molecules with a very short half-life and high reactivity. They can be oxygen-derived (ROS, reactive oxygen species), nitrogen-derived (RNS, reactive nitrogen species) or others (Figure 1) [41]. Several types of reactive species are generated in the body as a result of metabolic processes and the antioxidant system acts as an important counterbalance [44]. This system covers many enzymes (like superoxide dismutase, catalase or glutathione peroxidase), minerals, vitamins, glutathione, uric acid and others [44]. The central fact of oxidative stress is its double role: excessive oxidant challenge causes damage to biomolecules whereas a physiological level of oxidant challenge is essential for governing life processes through redox signaling [41]. In the case of cancer biology, an accelerated metabolism demands high ROS concentrations to maintain their high proliferation rate. Cancer cells develop different ways to increase ROS resistance including the execution of alternative pathways, which can avoid large amounts of ROS accumulation without compromising the energy demand [45]. Currently, the commonly used radio- and chemotherapeutic drugs influence tumor outcome through ROS modulation [45].

Oxidative stress as a mechanism of the cytotoxic activity of vanadium-based compounds is well documented. In one of the most recent studies, vanadium salts, sodium metavanadate NaVO_3_ (**36**) and vanadium(IV) sulfate oxideVOSO_4_ (**37**) significantly increased the ROS level in human lung cancer cells. However, the higher ROS level was induced by complexes containing vanadium(+IV) in the coordination center [46]. These results suggest that the efficacy of the ROS generation induced by vanadium compounds depends on the oxidation state of the vanadium cation presented in the coordination sphere of the complex [46]. For peroxovanadate complexes, the results are consistent. The polyacrylate derivative of peroxovanadate (**38**) inhibited growth of lung carcinoma cells by activating the axis of Rac1-NADPH oxidase leading to oxidative stress [47]. The vanadium complex with N-(2-hydroxyacetophenone) glycinate (**39**) triggered apoptosis in human colorectal carcinoma cells through mitochondrial outer membrane permeabilization, possibly by altering cellular redox status [48]. In the case of human pancreatic cancer cells, it has been shown that bis(acetylacetonato)-oxidovanadium(IV) complex (**40**) and sodium metavanadate (**36**) increased ROS generation; however, they did not induce a sustained increase of ROS generation, but the level of ROS reached a plateau instead [49]. Additionally, the results revealed that an intracellular feedback loop may be against the elevated ROS level, evidenced by the increased GSH content and the unchanged level of the antioxidant enzyme expression [49]. Oxidative stress was also induced in osteosarcoma cells by oxidovanadium(IV) complexes with glucose and naproxen (**41** and **42**) [50]. Leon et al. [51] have investigated the cytotoxicity of three oxidovanadium(IV) complexes (**43** and **45**) by use of the same type of cancer cells. The complex with phenanthroline (**45**) showed the highest cytotoxic activity which correlated with the strongest increase of ROS [51]. Interestingly, in studies performed by our group, the same vanadium complex (**45**–**47**) also induced the ROS generation in a human pancreatic cancer cell line [52]. In another study, an oxovanadium (IV/V) complex with a galactomannan derivative (polysaccharides) (**48** and **49**) showed cytotoxicity against a human liver cancer cell line by decreasing the mitochondrial membrane potential and increasing the ROS levels [53].

An interesting study has been performed by Li et al. [54] in which a vanadium dioxide nanocoating (VO_2_-modified) quartz surface has been prepared. The obtained results showed that the VO_2_-modified quartz surface (releasing ions from surface) interrupted the mitochondrial electron transport chain and then elevated the intracellular ROS levels in the cholangiocarcinoma cells [54].

An interesting and intriguing study performed by Wang et al. [55] suggests that vanadium complexes with antioxidants (**35, 40, 50**) should reduce their toxicities in human normal cells without affecting their antitumor activities in cancer cells (selective cytotoxicity). This is consistent with results for an oxidovanadium(IV) complex with 3-(3,4-Dihydroxycinnamoyl)quinic acid (**51**), which exhibited antioxidant activity as well as selective cytotoxicity against a human breast cancer cell line without increase of the ROS level [56]. The synthesis of vanadium complexes with flavonoids, well-known natural antioxidants [57,58], appears reasonable. Consequently, Naso et al. [59] have conducted a study on the oxidovanadium(IV) complex with flavonol morin (**52**). The new complex showed cytotoxic activity against breast cancer cell lines without generating reactive oxygen species in the cells and producing damage of DNA. Moreover, the complex did not affect the normal proliferation of the breast epithelial mammal cells [59]. These important results clearly demonstrate that the mechanism of cytotoxicity of vanadium compounds does not have to be ROS-dependent. On the other hand, many other flavonoid-based complexes showed the opposite mechanism. An oxidovanadium(IV) complex with apigenin (**53**) showed moderate cytotoxicity against lung and cervix cancer cell lines with simultaneous slight increments of ROS levels and decrease of the GSH/GSSG ratio [60]. Additionally, this cytotoxic activity was reverted when natural antioxidants were incubated with the complex [61]. In another study, an oxidovanadium(IV) complex with the flavonoid chrysin (**54**) caused a concentration-dependent inhibition of cell human osteosarcoma cells and ROS generation. The alterations in the GSH/GSSG ratio were proposed as the main mechanisms [61]. Besides, an oxidovanadium(IV) complex with flavonoid baicalin (**55**) also showed ROS-dependent cytotoxicity against a human lung cancer cell line [62].

Interestingly, an L-cysteine-based oxidovanadium(IV) complex (**56**) has been proposed as a promising chemoprotectant against oxidative stress and nephrotoxicity induced by cisplatin [63]. In this in vivo study, the vanadium complex exhibited strong nephroprotective efficacy by restoring antioxidant defense mechanisms [63]. A similar in vivo study has been performed for cyclophosphamide, that induces hepatotoxicity and genotoxicity in mice [64]. Oral administration of an L-cysteine-based oxidovanadium(IV) complex (**56**) significantly attenuated cyclophosphamide-induced oxidative stress in the liver as evident from levels of reactive oxygen species, nitric oxide and lipid peroxidation [64]. In addition, it restored the glutathione level and activities of antioxidant enzymes, and mitigated chromosomal aberrations, micronuclei formation, DNA fragmentation and apoptosis in bone marrow cells and DNA damage in lymphocytes [64].

In case of photodynamic therapy (PDT) (described in the DNA subsection), induction of oxidative stress has been documented. Ferrocenyl-terpyridine oxidovanadium(IV) complexes (**57**–**60**) exhibit photocytotoxic activity against breast and cervical cancer cells through ROS generation [65]. Moreover, investigated complexes show significant photocleavage of plasmid DNA in green light forming ·OH radicals [65]. The structures and activities of ROS-inducing vanadium compounds are summarized in Table 2.

The free radical generation may be associated with DNA damage. It has been determined that vanadium(IV) (**37**) caused molecular oxygen-dependent DNA strand breaks as well as molecular oxygen dependent 2′-deoxyguanosine (dG) hydroxylation to form 8-hydroxyl-2′-deoxyguanosine (8-OHdG) [66]. Moreover, incubation of VOSO_4_ (**37**) with dG in argon did not generate any significant amount of 8-OHdG [66]. For some vanadium-based complexes described above, DNA damage with simultaneous ROS generation was also recognized [26,27,36,37,47,48] (Table 1 and Table 2). DNA lesions lead to cell cycle disruption and ultimately to cell death, which are discussed in the following subsections.

### 2.3. Cell Cycle Arrest

The cell cycle is the sequence of stages through which a cell passes between one cell division and the next. This process includes four stages: G_1_, S, G_2_ and M phases [67]. In S phase, the genetic material is replicated (DNA synthesis), whereas M phase includes mitosis and cytokinesis. G_1_ and G_2_ are “gaps” during which time cells prepare for the next phase [67]. The passage to the different phases is coordinated by a set of the proteins: cyclins and their associated cyclin-dependent kinases (cdks) [68]. Cyclin/cdk complexes play a key role in checkpoints, which monitor progression through each cell cycle phase and maintain the correct order of events [68]. If aberrant or incomplete cell cycle events (e.g., DNA damage) are detected, checkpoint pathways trigger cell cycle arrest until the problem is resolved [68]. During cell cycle arrest, cells can repair cellular damage, spread an exogenous cellular stress signal or increase availability of essential growth factors, hormones, or nutrients [68]. The system of cell cycle regulation also includes the CDK inhibitors (CDKIs), which are divided into two families: Ink4 family (p16, p15, p18 and p19) and the Cip/Kip family (p21, p27, p57) [69]. Moreover, the p53 protein also plays an important role in cell cycle regulation. It has been shown that p53 and p21 are necessary to maintain a G_2_ arrest following DNA damage [70,71]. The cdc25 phosphatase family is another group which activates cyclin-dependent kinases through dephosphorylation [72]. Deregulation of the cell cycle characterizes cancer cells, which underlies the aberrant cell proliferation and promotes genetic instability [67].

The impact of vanadium-based compounds on cell cycle progression was determined and described earlier [73]. More recent research is coherent with previous studies and supplements our knowledge in this area. Liu et al. [74] have shown that sodium metavanadate (**36**) caused G_2_/M cell cycle arrest in prostate cancer cells, which is evidenced by the increase in the level of phosphorylated cdc2(cdk1) at its inactive Tyr-15 site. Importantly, the results revealed that ROS-mediated degradation of cdc25C is responsible for vanadate-induced G_2_/M cell cycle arrest [74]. Furthermore, sodium orthovanadate (**61**) induced G_2_/M phase cell cycle arrest in a human thyroid carcinoma cell line [75]. Interestingly, the same vanadium salt (**61**) decreased the expression of cyclin D1 and increased the expression of p21 protein in papillary thyroid carcinoma-derived cells; however, the cell cycle profile was similar to the untreated cells [76]. In other study in malignant melanoma cell lines, the inorganic anion vanadate(V) (**62**) arrested the cell cycle in G_2_/M phase whereas pirydone-based vanadium complex (**63**–**65**) arrested it in the G_0_/G_1_ phase [77]. Furthermore, both compounds (**62** and **63**) induced dephosphorylation of the Retinoblastoma protein (Rb) and together had a pronounced increase of cyclin-dependent kinase inhibitor p21 protein expression [78]. These studies highlight the importance of the chemical form of vanadium-based complexes in determining their mechanism of action. In our team study, the oxidovanadium complex containing quinolinium cation (**66**) induced cycle arrest in the G_2_/M phase with simultaneous triggering of the p53/p21 pathway in pancreatic cancer cell lines [79]. Induction of the p53/p21 pathway was also determined in cervical cancer cells treated by vanadium complexes of nicotinoyl hydrazine (**67** and **68**) [80]. An oxidovanadium complex with phenanthroline (**69**–**72**) arrested the cell cycle in the S and G_2_/M phases in hepatocellular carcinoma cell lines [81]. Contrary to this study, other phenanthroline-based vanadium complex (**73** and **74**) caused a G_0_/G_1_ phase cell cycle arrest in the same type of cancer cell lines [82]. Additionally, G_0_/G_1_ phase cell cycle arrest, induced by organic vanadium complexes (**75**–**78**), was shown in human neuroblastoma cells [83]. Cell cycle arrest in S phase has been determined in esophageal squamous carcinoma cell lines treated by sodium vanadate (**61**) [84]. Additionally, diaminotris(phenolato) vanadium(V) complexes (**79** and **80**) arrested the cell cycle at the S phase in human colon cancer and ovarian carcinoma cell lines [85]. All above studies clearly suggest that the mechanism of cell cycle disruption depends not only on organic ligands and the spatial structure of vanadium-based compounds but also on the type of cancer cell lines, which may be associated with their genetic background. The structures and activity of cell-cycle-disrupting vanadium compounds are summarized in Table 3.

Some studies, described in the previous subsection, have found a connection between DNA binding and cell cycle arrest [27,28,31,40] (Table 1) as well as oxidative stress and cell cycle progression [49,52,55] (Table 2). Wu et al. [49] have demonstrated that the ROS-induced sustained MAPK/ERK activation contributed to vanadium-compound-induced G_2_/M cell cycle arrest in pancreatic cancer cells. Additionally, cell cycle disruption with simultaneous ROS generation was shown in lung carcinoma [47] and osteosarcoma cell lines [61].

### 2.4. Programed Cell Death

Apoptosis is programed cell death with distinct genetic and biochemical pathways that play a critical role in development and homeostasis in normal tissues [86]. Apoptosis is caused by special proteases, caspases, which specifically target cysteine aspartyl [86]. Moreover, there are two main pathways to apoptotic cell death: the extrinsic pathway mediated by membrane death receptors and the intrinsic pathway mediated by the mitochondria [86]. Under many stressful conditions, like activation of the DNA damage checkpoint pathway, apoptosis can remove potentially harmful DNA-damaged cells in order to block carcinogenesis [87]. Defects in this process can cause cancer. The cancer cells use some of several molecular mechanisms to suppress apoptosis and acquire resistance to apoptotic agents including the downregulation or mutation of proapoptotic proteins such as BAX expression or the expression of antiapoptotic proteins such as Bcl-2 [87].

Induction of apoptotic cell death by vanadium-based compounds is well established. Sodium orthovanadate (Na_3_VO_4_) (**61**) induced apoptosis in the oral squamous cell carcinoma cell line [88] as well as in human anaplastic thyroid carcinoma cells [75]. Moreover, orthovanadate (**61**) induced typical features of apoptosis including DNA fragmentation, loss of mitochondrial membrane potential and activation of caspase-3 in thyroid cancer cells harboring RET/PTC1 (oncogenic chromosomal rearrangements) [76].

In the case of some vanadium-based complexes, it has been exhibited that vanadium complex (**81**) induced apoptosis in gastric cancer lines via the mediation of the intrinsic apoptotic pathway (upregulation of Bax, PARP and caspase-3/9) [89]. Moreover, vanadium complex with the flavonoid quercetin (**82**) upregulated the expressions of p53 and caspase 3 and 9 with simultaneous downregulation of Akt, mTOR and VEGF expressions in human breast cancer cell lines [90]. Additionally, complexes with flavonoid quercetin (**82**) induced apoptosis in a breast cancer animal model [90]. On the contrary, a vanadium complex containing phenanthroline (**83**) triggered apoptosis by activation of both extracellular (through caspase 8) and intracellular (through caspase 9) apoptosis-inducing pathways leading to activation of downstream caspase 3 in the human T-leukemic cells [91]. A vanadium complex with a modified phenol group (**84**) induced apoptosis in liver hepatocellular carcinoma cells using the p53-p21 pathway-dependent way [92]. A p53-dependent apoptotic mechanism, induced by vanadium complexes of nicotinoyl hydrazine (**67** and **68**), was also determined in cervical cancer cells [80]. A complex study, using the functional proteomic analysis, has been performed on human osteosarcoma cells. The results showed that an oxidovanadium(IV) complex with the clioquinol (**85**) induced upregulation of proteins such as caspase 3, caspase 6, caspase 7, caspase 10, caspase 11, Bcl-x and DAPK, as well as downregulation of ones such as PKB/AKT and DIABLO [93]. Moreover, cell signaling pathways involved in several altered pathways related to the PKC and AP2 family were identified [93]. An interesting study has been performed using photodynamic therapy. Oxidovanadium(IV) vitamin-B6 Schiff base complexes (**86** and **87**) showed remarkable apoptotic photocytotoxicity in visible light and specific localization to endoplasmic reticulum (ER) in ovarian and breast cancer cell lines [94].

Induction of oxidative stress may lead to apoptotic cell death [95]. Vanadium inorganic salts, namely NaVO_3_ (**36**) and VOSO_4_ (**37**), exhibited apoptotic-inducing cytotoxicity against non-small lung cell carcinoma cells with simultaneous increase of ROS level [46]. Pisano et al. have shown that both the inorganic anion vanadate(V) (**62**) and the vanadium complex with pyridinonate (**63**) induced apoptosis through generation of ROS in malignant melanoma cells [78]. Interestingly, a vanadium-Schiff base complex (**39**) actuated apoptosis through mitochondrial outer membrane permeabilization in human colorectal carcinoma cells; however, this was in a caspase-independent manner, possibly by altering cellular redox status and inflicting DNA damage [48].

Evading programed cell death is one of the hallmarks of cancer [96]. Therefore, seeking an alternative nonapoptotic form of programed cell death is required [97,98,99]. It has been found that dioxovanadium complexes with substituted salicylaldehyde derivatives (**88** and **89**) induced cell death in colon cancer cells via the activation of RIPK3 and necroptosis pathway [100]. Furthermore, we have determined that vanadium complexes with phenanthroline (**45**, **47**) also trigger the necroptosis pathway in a human pancreatic ductal adenocarcinoma cell line [52]. In another study, we have found that an oxidovanadium complex with quinolinium cation (**66**) induced autophagic process in pancreatic cancer cells with simultaneous increase in the RAGE protein level [79]. Autophagy was also detected in vanadium-treated (**90**) breast cancer cells [101]. An interesting study has been performed in hypoxic conditions. A complex of the hydrolysate of galactomannan with oxidovanadium(IV) (**91**) exhibited strong apoptotic activity against hepatocellular carcinoma cell lines under normoxic conditions. However, this was completely lost under hypoxic conditions [102]. This was explained by strong induction of autophagy, which was characterized as a pro-survival mechanism in hypoxia [102]. Autophagy may play a dual role in cancer cells and therefore both its induction and inhibition can provide a valuable therapeutic strategy [99]. Structures and mechanisms of cell death induced by vanadium compounds are summarized in Table 4. Many DNA-binding, ROS-inducing and cell-cycle-disrupting vanadium compounds induce programed cell death and are described in Table 1, Table 2 and Table 4.

Summary of the described molecular and cellular mechanisms of vanadium compounds are illustrated in Figure 2.

### 2.5. Other Mechanims

Many other mechanisms in the anticancer activity of vanadium compounds have been determined which are described in other reviews.

Kioseoglou et al. [20] described impacts on cell metabolism. For example, the level of mRNA of key metabolicglycolytic enzymes in the liver, including phosphoenolpyruvate carboxykinase (PEPCK), glucokinase (GK), and L-pyruvate kinase (L-PK), were significantly restored toward normal values in diabetic animals treated with vanadium compounds [103]. Moreover, sodium vanadate affected the activity of pulmonary 6-phosphogluconate dehydrogenase (6PGDH), pyruvate kinase (PK) and glutathione peroxidase (GP), and in higher doses lactate dehydrogenase (LDH) and glutathione reductase (GR) [104]. Because cancer cells exhibit drastically enhanced glucose uptake and glycolysis (the Warburg effect), vanadium-based compounds could be an interesting treatment option [20]. In the same review, the molecular mechanism of the epithelial–mesenchymal transition (EMT) inhibition by vanadium was described [20]. EMT is a process during which epithelial cells lose their polarized organization and cell-to-cell adhesion and undergo changes in cell shape and cytoskeletal organization, which ultimately lead to cell migration and invasion [20].

Irving and et al. [105] discussed the potential of vanadium derivatives as protein tyrosine phosphatases (PTP) enzyme inhibitors. Phosphotyrosine signaling is implicated in almost all aspects of cancer biology due to its widespread influence over cell signaling, and alterations brought about by mutations can drive the initiation and progression of many different tumor types [106]. It has been determined that various vanadium compounds were shown to be successful in inhibiting tumor development in animal models through their ability to inhibit PTPs and to induce oxidative damage, which itself likely contributes to PTP inhibition [105].

Interestingly, both pro- and anti-inflammatory effects have been documented for V-containing compounds [107]. The immunomodulatory activity of vanadium compounds includes effects on T cell, B cell and NK cell activity as well as effects on the level of proinflammatory cytokines and mediators such as NF-κB, COX-2 or IL-6 [107]. For instance, NH_4_VO_3_ was exhibited to prevent T cell activation by downregulating the expression of proinflammatory cytokines including IL-2, IL-6, TNF-α, and IFN-γ [108]. Triggering Toll-like receptors (TLR) to generate an immune response is considered to be another mechanism through which vanadium compounds could regulate immune response [107].

Furthermore, in addition to potential anticancer activity, vanadium compounds exhibit a well-established antidiabetic activity [109]. Vanadate binds to the active side of PTP-1B (which counteracts the insulin receptor (IR) in the absence of insulin or in the insufficient insulin response), due to its similarity to phosphate, and inhibits it. Consequently, signal transduction paths for glucose uptake are restored [109].

## 3. Conclusions

The studies described in this review suggest that the molecular and cellular mechanisms of vanadium compounds depend on many factors, including the oxidation state of the vanadium cation, organic ligands, spatial structure and also the type of cancer cell lines. That is why we can observe so many, sometimes mutually exclusive, mechanisms of cytotoxicity.

The literature review revealed that the chemical form of vanadium-based complexes (oxidation state of vanadium, the type of the ligands and their geometrical arrangement) influences their physicochemical properties and thus their biological properties. For instance, the presence of a strong binding ligand in the coordination sphere of the VO^2+^ ion hinders oxidation of the metal ion, V(IV) to V(V) [110]. Furthermore, nuclease activity of the V-phenanthroline, V-bipyridine and V-terpyridine compounds depends on the number of intercalating heterocyclic moieties. It suggests that the incorporation into the coordination sphere of the vanadium cation of the appropriate type of ligands may promote redox reactions or enhance the interaction with nucleic acids. This leads to oxidative stress, DNA damage, cell cycle arrest and ultimately to cell death (Figure 2).

Although there is some evidence that the structure and physicochemical properties of vanadium complexes impact their biological activity, the correlation of the chemical form of vanadium-based complexes versus their mechanism of action still remains to be elucidated. Moreover, the differences in physicochemical and biological properties of the compounds may stem from very different experimental (chemical and biological) conditions. The results of chemical studies on physicochemical properties of the compounds should be assessed very carefully as the experimental conditions leading to these results are generally very different from biological conditions. Furthermore, there are still very few in vivo studies and an almost complete lack of innovative approaches based on targeted therapies. In view of this, further biological research, focusing on a more in-depth analysis of cytotoxic activity using the most modern techniques, is required.

In conclusion, the above considerations underline the anticancer potential of vanadium-based compounds. Through modification of the chemical form of vanadium-based complexes, we can influence affinity for DNA, oxidative stress or the type of cell death induced by vanadium-based compounds in cancer cells. On the other hand, due to many factors, it is difficult to precisely define structure–activity relationships.

## Figures and Tables

**Figure 1 molecules-25-01757-f001:**
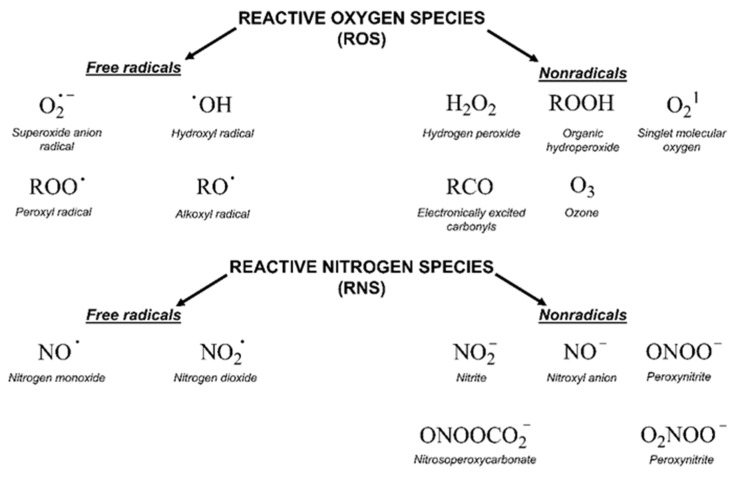
Chemical structure and nature of reactive oxygen (ROS) and nitrogen species (RNS).

**Figure 2 molecules-25-01757-f002:**
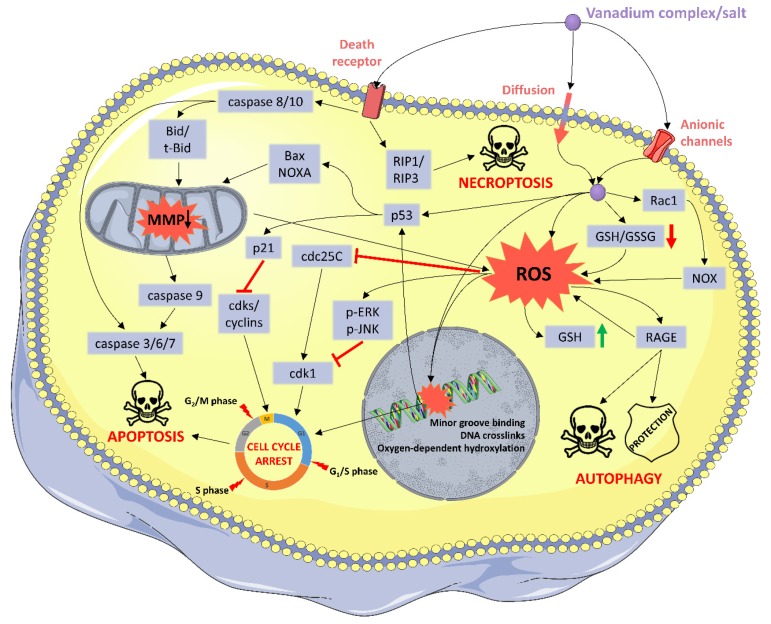
Summary of the suggested molecular and cellular mechanisms of vanadium compounds. ROS: reactive oxygen species; MMP: mitochondrial membrane potential; GSH: reduced glutathione; GSSG: oxidized glutathione; NOX: NADPH oxidase. Elements of this illustration were provided by Servier Medical Art (http://smart.servier.com/).

**Table 1 molecules-25-01757-t001:** Structures and mechanism of action DNA-binding vanadium compounds (K_b_-binding constant).

Structure	Activity	References
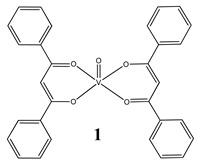	Groove binding to salmon sperm DNA accompanied with a partial insertion between the base stacks of the DNA (**K_b_ = 2.3 × 10^3^ M^−1^**) Cytotoxicity (24 h): breast cancer cells MCF-7 (**IC_50_ 7.8 µM**) liver cancer cells HepG2 (**IC_50_ 13.5 µM**) colon cancer cells HT-29 (**IC_50_ 16.1 µM**)	[25]
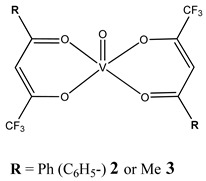	Oxidative cleavage of DNA through the generation of a hydroxyl radical Minor groove binding to DNA (**2: K_b_ = 1.95 ± 0.16 × 10^3^ M^−1^** **3: K_b_ = 1.064 ± 0.17 × 10^3^ M^−1^**) Cytotoxicity (48 h): cervical cancer cells HeLa (**2: IC_50_ 256.9 µM 3: IC_50_ 480.5 µM**)	[26]
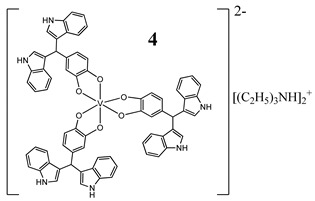	Similarities to cisplatin concerning DNA interaction ROS generation, mitochondrial damage, G_2_/M cell cycle arrest Cytotoxicity (72 h): panel of melanoma, colon, cervical, breast and pancreatic cancer cells **IC_50_ < 10 µM for all cell lines**	[27]
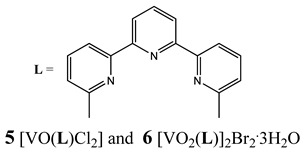	Intercalation as the way of DNA binding G_2_/M cell cycle arrest Cytotoxicity (48 h): cervical cancer cells HeLa (**5: IC_50_ 42.9 ± 1.5 µM** **6: IC_50_ 33.2 ± 0.9 µM**) breast cancer cells T-47D (**5: IC_50_ 38.0 ± 1.6 µM** **6: IC_50_ 42.3 ± 1.8 µM**) Lung cancer cells A549 (**5: IC_50_ 87.6 ± 2.4 µM** **6: IC_50_ > 100 µM**)	[28]
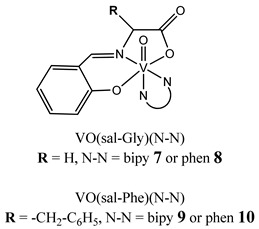	Phen-containing V^IV^O compounds display stronger DNA interaction ability than the corresponding bipy analogues Cytotoxicity (72 h): ovarian cancer cells A2780 (**7: IC_50_ 20.8 ± 0.5 µM 8: IC_50_ 4.9 ± 1.3 µM** **9: IC_50_ 17.1 ± 3.9 µM 10: IC_50_ 4.7 ± 1.8 µM**) breast cancer cells MCF-7 (**7: IC_50_ 53 ± 2.0 µM 8: IC_50_ 77 ± 1.3 µM** **9: IC_50_ 95 ± 3.7 µM 10: IC_50_ 68 ± 1.4 µM**)	[29]
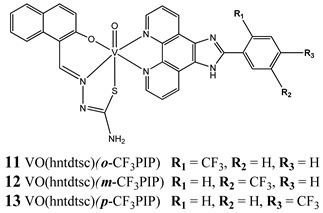	Interaction with CT-DNA through a non-classical intercalative mode cleavage plasmid pBR322 DNA upon exposure to ultraviolet light Cytotoxicity (48 h): panel of cervical, breast and esophageal cancer cells **IC_50_ range: 0.31–6.15 μM**	[30]
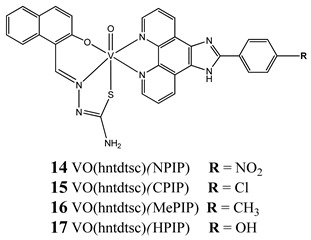	Binding with CT-DNA by an intercalation **K_b_ = 14: 1.53 × 10^5^ M^−1^ 15: 1.41 × 10^5^ M^−1^** **16: 1.05 × 10^5^ M^−1^ 17: 0.95 × 10^5^ M^−1^** cleave supercoiled plasmid DNA in the presence of H_2_O_2_ G_0_/G_1_ cell cycle arrest (**14**) Induction apoptosis in Hela cells (**14**) Cytotoxicity (24 h): cervical cancer cells HeLa (**14: IC_50_ 1.09 ± 0.16 µM** **15: IC_50_ 10.36 ± 1.23 µM**) bladder cancer cell BIU-87 (**14: IC_50_ 4.51 ± 0.68 µM** **15: IC_50_ 8.69 ± 1.05 µM**) lung cancer cells SPC-A-1 (**14: IC_50_ 7.61 ± 0.55 µM** **15: IC_50_ 21.43 ± 3.24 µM**)	[31]
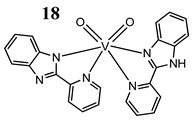	Interaction with DNA in a intercalative fashion (**K_b_ = 2.76 × 10^5^ M^−1^**) Cytotoxicity (24 h): lung cancer cell A549 breast cancer cells MCF-7 keratinocyte cancer cell A431 **IC_50_ for all cancer cell lines 75 μM** normal human keratinocyte cells HaCaT **IC_50_ 150 µM**	[32]
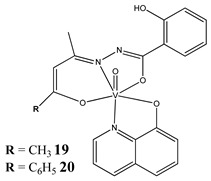	The intercalative mode of binding to DNA (**19: K_b_ = 6.13 × 10^5^ M^−1^** **20: K_b_ = 8.69 × 10^5^ M^−1^**) Cytotoxicity (24 h): cervical cancer cell SiHa (**19: IC_50_ 33 µM 20: IC_50_ 29 µM**)	[33]
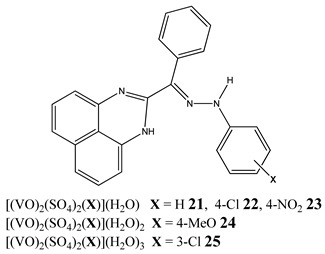	Binding to CT-DNA **K_b_ = 21: 6.10 × 10^4^ M^−1^ 22: 7.99 × 10^4^ M^−1^** **23: 6.75 × 10^4^ M^−1^ 24: 6.07 × 10^4^ M^−1^** **25: 8.80 × 10^4^ M^−1^** Cytotoxicity (48 h): breast cancer cells MCF-7 (**25: IC_50_ 11.44 µM 23: IC_50_ 15.50 µM**) liver cancer cells HepG2 (**25: IC_50_ 9.91 µM 23: IC_50_ 11.01 µM**) colon cancer cells HCT 116 (**24: IC_50_ 13.27 µM 23: IC_50_ 15.53 µM**)	[34]
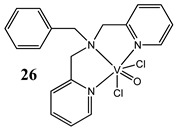 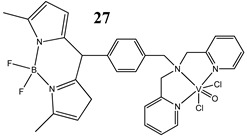	Light-activated VO^2+^-DNA crosslink formation (**27**) singlet oxygen (^1^O_2_) induced mitochondria-targeted PDT (**27**) Cytotoxicity (24 h): breast cancer cells MCF-7 (**27: IC_50_ 3.4±0.4 µM** in visible light **IC_50_ > 50 µM** in the dark) cervical cancer cells HeLa (**27: IC_50_ 1.8±0.6 µM** in visible light **IC_50_ > 50 µM** in the dark) **26**: any significant cytotoxicity in light	[36]
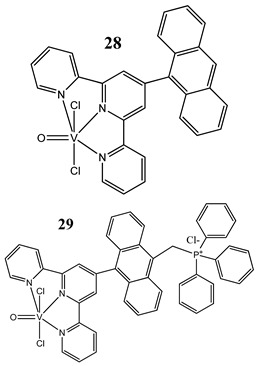	Light-activated DNA crosslink formation (in the dark they are partial DNA intercalators) ROS generation in visible light Cytotoxicity (24 h): breast cancer cells MCF-7 (**28: IC_50_ 10.4 ± 1.6 µM** in visible light **IC_50_ > 50 µM** in the dark) (**29: IC_50_ 2.3 ± 0.3 µM** in visible light **IC_50_ 27.6 ± 1.4 µM** in the dark) cervical cancer cells HeLa (**28: IC_50_ 8.2 ± 0.3 µM** in visible light **IC_50_ > 50 µM** in the dark) (**29: IC_50_ 1.8 ± 0.5 µM** in visible light **IC_50_ 20.3 ± 1.0 µM** in the dark)	[37]
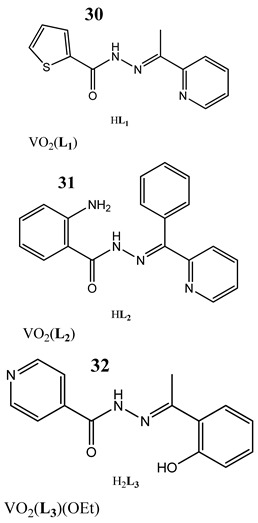 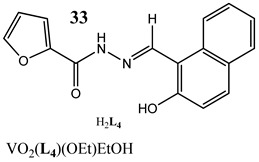	Photo-induced cleavage of pUC19 supercoiled plasmid DNA Interaction with CT-DNA through minor groove binding mode **K_b_ = 30: 8.56 × 10^4^ M^−1^ 31: 1.13 × 10^5^ M^−1^** **32: 4.95 × 10^4^ M^−1^ 33: 5.03 × 10^3^ M^−1^** Cytotoxicity (72 h): cervical cancer cells HeLa **30: IC_50_ 20 ± 4.52 µM** **31: IC_50_ 18 ± 3.38 µM** **32: IC_50_ 19.5 ± 3.54 µM** **33: IC_50_ 9.9 ± 3.18 µM**	[38]
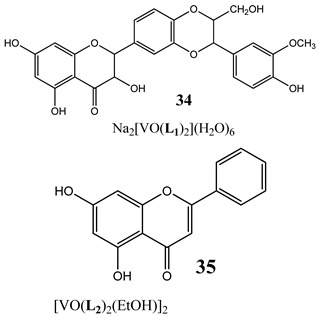	The topoisomerase IB inhibition (**34**) G_2_/M cell cycle arrest (**35**) activation caspase 3 and triggering the apoptosis (**34**) Cytotoxicity (24 h): colon cancer cells HT-29 A concentration-related inhibition **from 75 to 100 µM**	[40]

Bold and Underline: makes Table more readable.

**Table 2 molecules-25-01757-t002:** Structures and mechanism of action ROS-inducing vanadium compounds (ROS, reactive oxygen species; MMP, mitochondrial membrane potential; GSH/GSSG, reduced/oxidized glutathione).

Structure	Activity	References
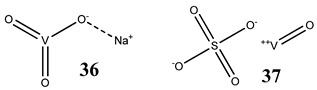	Lung cancer cells A549 12(**36**)- and 14(**37**)-fold increase in ROS generation (100 μM after 48 h) Cytotoxicity: 24 h (**36**: **IC_50_** >**100 µM** **37**: **IC_50_** >**100 µM**) 48 h (**36**: **IC_50_** ~**100 µM** **37**: **IC_50_** >**100 µM**)	[46]
[V_2_O_2_(O_2_)4(carboxylate)]-**PA** 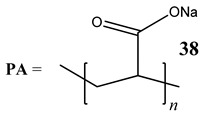	Lung cancer cells A549 Activation of the axis of Rac1-NADPH oxidase leading to oxidative stress Increase in phosphorylation of H2AX (γH2AX), a marker of DNA damage	[47]
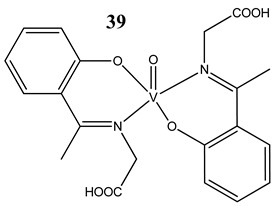	Breast cancer cells MCF-7 Glioblastoma cells U-373MG T lymphoblastic leukaemia cells CCRF-CEM and CEM-ADR 5000 Colon cancer cells HCT-116 Depletion of GSH content ROS generation Induction of apoptosis through mitochondrial outer membrane permeabilization but in caspase independent manner	[48]
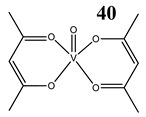	Pancreatic cancer cells AsPC-1 ROS generation G_2_/M cell cycle arrest Activation of PI3K/AKT and MAPK/ERK signaling pathways Increased level of phosphorylated Cdc2 at Tyr-15 and the reduced level of Cdc25C	[49]
GluVO **41** NapVO **42** 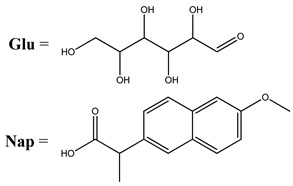	Osteosarcoma cells UMR106 Induction of apoptosis ROS generation	[50]
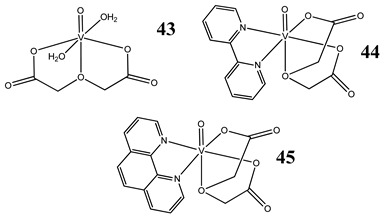	Osteosarcoma cells MG-63 Induction of oxidative stress, apoptosis and DNA cleavage Cytotoxicity (24 h): **43: IC_50_ >100 µM** **44: IC_50_ >100 µM** **45: IC_50_ 58 µM**	[51]
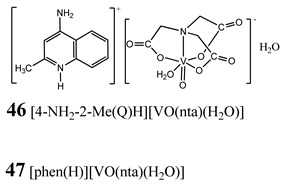	Pancreatic cancer cells PANC-1 3-fold increase in ROS generation (**45**,**47**) (5 µM 48 h) 8-fold increase in ROS generation (**46**) (10 µM 48 h) G_2_/M cell cycle arrest (**46**) Induction of necroptosis (**45**, **47**) Induction of autophagy (**46**) Cytotoxicity (48 h): **45: logIC_50_ 0.52 ± 0.28 µM** (**IC_50_ 3.3 µM**) **46: logIC_50_ 1.47 ± 0.07 µM** (**IC_50_ 29.5 µM**) **47: logIC_50_ 1.10 ± 0.11 µM** (**IC_50_ 12.6 µM**)	[52]
**48** SAGM:VO SAGM = native galactomannan from S. amazonicum **49** MSAGM:VO MSAGM = modified form native galactomannan from S. amazonicum	Liver cancer cells HepG2 ROS generation Reduction in MMP	[53]
VO_2_-modified quartz surface (releasing ions from surface)	Cholangiocarcinoma cells ROS generation Interruption of the mitochondrial electron transport chain	[54]
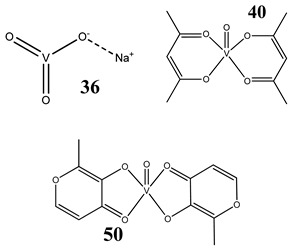	Liver cancer cells HepG2 Insignificant ROS generation G_1_/S cell cycle arrest Immortalized hepatic cells L02 cells ROS generationS and G_2_/M cell cycle arrest	[55]
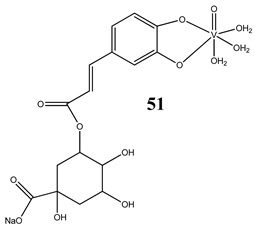	Antioxidant activity (inhibitory effects on O_2_^·−^, ^·^OH and ROO^·^ radicals generation) Selective cytotoxicity against breast cancer cells SKBR3 No increase in ROS generation	[56]
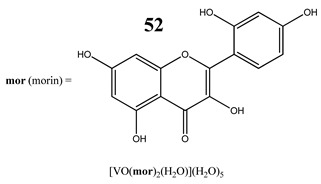	Breast cancer cells SKBR3 and T47D Apoptotic cell death process with caspase 3/7 activation Perturbation of the MMP No increase in ROS generation No effect on the normal proliferation of the breast epithelial mammal cells	[59]
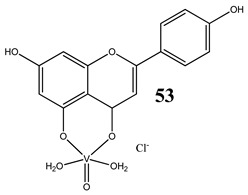	Lung cancer cells A549 Cervical cancer cells HeLa ROS generation GSH/GSSH depletion Cytotoxicity: Lung cancer cells A549 24 h (**IC_50_ >100 µM**) 48 h (**IC_50_ 17.7 ± 3.1 µM**) 72 h (**IC_50_ 2.2 ± 0.9 µM**) Cervical cancer cells HeLa 24 h (**IC_50_ 88.1 ± 1.4 µM**) 48 h (**IC_50_ 115.5 ± 1.6 µM**) 72 h (**IC_50_ 9.7 ± 1.9 µM**)	[60]
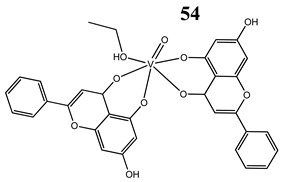	Osteosarcoma cells MG-63 ROS generation GSH/GSSH depletion Induction of apoptosis (increased levels of caspase 3) Disruption of the MMP	[61]
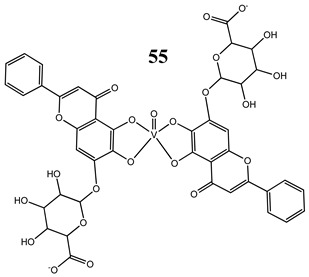	Lung cancer cells A549 ROS generation Cytotoxicity: 24 h (**IC_50_ >100 µM**) 48 h (**IC_50_ 44.7 ± 3.5 µM**) 72 h (**IC_50_ 21.7 ± 1.3 µM**)	[62]
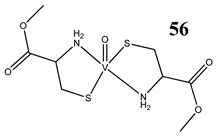	Swiss albino mice Nephroprotective efficacy (against cisplatin) by restoring antioxidant defense mechanism	[63]
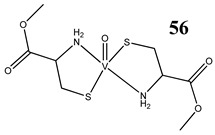	Swiss albino mice Prevention of cyclophosphamide-induced hepatotoxicity and genotoxicity Restoration of glutathione level and activities of antioxidant enzymes	[64]
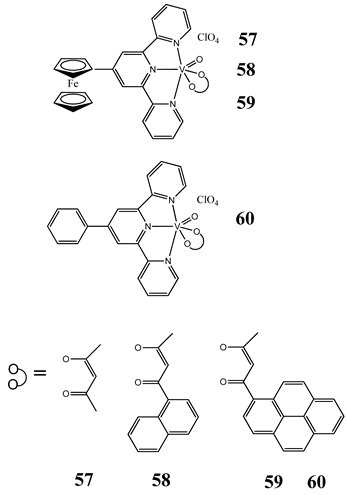	Cervical cancer cells HeLa Breast cancer cells MCF-7 ROS generation Photocleavage of plasmid DNA in green light (568 nm) forming ^·^OH radicals Low toxicity in normal fibroblast 3T3 cells Cytotoxicity (24 h): cervical cancer cells HeLa (**58: IC_50_ 15.5 ± 1.0 µM** in visible light **IC_50_ 45.1 ± 1.2 µM** in the dark) (**59: IC_50_ 5.4 ± 0.5 µM** in visible light **IC_50_ 37.6 ± 1.2 µM** in the dark) (**60: IC_50_ 19.9 ± 1.1 µM** in visible light **IC_50_ 40.1 ± 1.1 µM** in the dark) breast cancer cells MCF-7 (**57: IC_50_ 5.6 ±0.9 µM** in visible light **IC_50_ 41.2 ± 1.0 µM** in the dark) (**58: IC_50_ 3.3 ± 0.6 µM** in visible light **IC_50_ 48.7 ± 1.1 µM** in the dark) (**59: IC_50_ 2.1 ± 0.3 µM** in visible light **IC_50_ 38.5 ± 1.1 µM** in the dark)	[65]

Bold and Underline: makes Table more readable.

**Table 3 molecules-25-01757-t003:** Structures and mechanism of action of cell-cycle-disrupting vanadium compounds (ROS, reactive oxygen species; MMP, mitochondrial membrane potential).

Structure	Activity	References
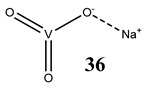	Prostate cancer cells PC-3 G_2_/M cell cycle arrest ROS-mediated degradation of Cdc25C Increase in the level of phosphorylated Cdc2 at its inactive Tyr-15 site	[74]
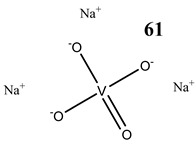	Thyroid carcinoma cells 8505C G_2_/M cell cycle arrest Induction of apoptosis Reduction in MMP Cytotoxicity: 24 h (**IC_50_ 3.76 µM**) 48 h (**IC_50_ 3.55 µM**) 72 h (**IC_50_ 3.23 µM**) 96 h (**IC_50_ 1.62 µM**)	[75]
	Papillary thyroid carcinoma-derived cells TPC-1 Decrease in the expression of cyclin D1 Increase in the expression of p21 Undisturbed cell cycle ROS generation Induction of apoptosis (activation of caspase-3) Reduction in MMP Increase in phosphorylation of tyrosine 451 of RET/PTC1 and activation of the mTOR/S6R branch of the PI3K/Akt signaling pathway	[76]
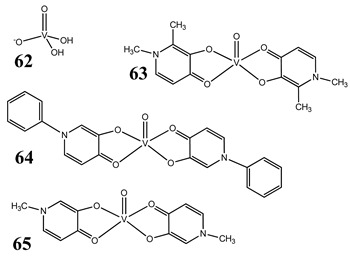	Malignant melanoma cells A375 and CN-mel Induction of apoptosis G_2_/M cell cycle arrest (**62**) G_1_/S cell cycle arrest (**63**) Cytotoxicity (72 h): A375 cells **62: IC_50_ 4.7 µM 63: IC_50_ 2.6 µM** **64: IC_50_ 4.2 µM 65: IC_50_ 2.4 µM** CN-mel cells **62: IC_50_ 6.5 µM 63: IC_50_ 12.4 µM** **64: IC_50_ 14.0 µM 65: IC_50_ 10.4 µM**	[77]
	Malignant melanoma cells A375 ROS generation (**62**, **63**) Induction of dephosphorylation of Rb protein (**62**, **63**) Cell cycle arrest by contrasting MAPK pathway activation and strongly inducing p21 expression and Rb hypophosphorylation (**62**, **63**)	[78]
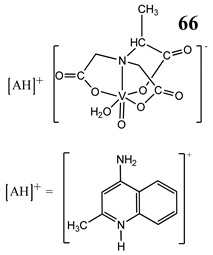	Pancreatic cancer cells PANC-1 and MIA PaCa2 G_2_/M cell cycle arrest (increase in the expression of cyclinB1 and cdk1) Induction of p53/p21 pathway Induction of autophagy Increase in the expression of RAGE ROS generation Cytotoxicity (48 h): PANC-1 cells: **IC_50_ 44.67 µM** MIA PaCa2 cells: **IC_50_ 72.22 µM**	[79]
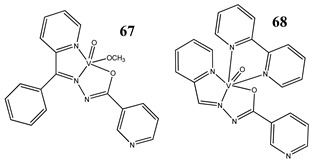	Cervical cancer cells HeLa and SiHa Induction of p53/p21 pathway Induction of apoptosis	[80]
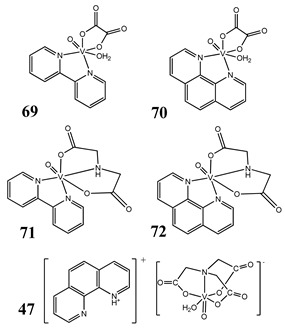	Liver cancer cells SMMC-7721, HepG2 S and G_2_/M cell cycle arrest (**70**) Induction of apoptosis (**70**) Cytotoxicity (48 h): SMMC-7721 cells **69: IC_50_ 60.19 ± 0.03 µM** **70: IC_50_ 5.34 ± 0.03 µM** **71: IC_50_ 119.44 ± 0.03 µM** **72: IC_50_ 25.55 ± 0.02 µM** **47: IC_50_ 42.46 ± 0.03 µM** HepG2 cells **69: IC_50_ 52.33 ± 0.02 µM** **70: IC_50_ 29.07 ± 0.01 µM** **71: IC_50_ 106.13 ± 0.02 µM** **72: IC_50_ 39.63 ± 0.03 µM** **47: IC_50_ 101.62 ± 0.02 µM**	[81]
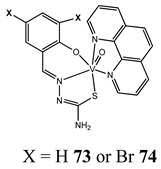	Liver cancer cells BEL-7402, HUH-7 and HepG2 G_1_/S cell cycle arrest Reduction in MMP Induction of apoptosis	[82]
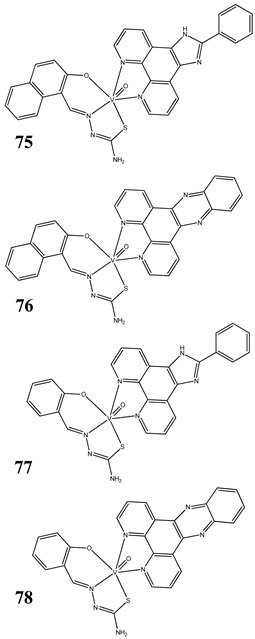	Neuroblastoma cells SH-SY5Y and SK-N-SH Breast cancer cells MCF-7 G_1_/S cell cycle arrest (**75**) Induction of apoptosis (**75**) Cleavage plasmid pBR322 DNA Cytotoxicity (48 h): SH-SY5Y cells **75: IC_50_ 1.08 ± 0.26 µM** **76: IC_50_ 2.30 ± 0.09 µM** **77: IC_50_ 1.69 ± 0.21 µM** **78: IC_50_ 3.92 ± 0.43 µM** SK-N-SH cells **75: IC_50_ 0.21 ± 0.023 µM** **76: IC_50_ 0.48 ± 0.017 µM** **77: IC_50_ 0.37 ± 0.025 µM** **78: IC_50_ 1.42 ± 0.11 µM** MCF-7 cells **75: IC_50_ 6.49 ± 0.28 µM** **76: IC_50_ 4.41 ± 1.27 µM** **77: IC_50_ 8.99 ± 0.39 µM** **78: IC_50_ 8.63 ± 1.31 µM**	[83]
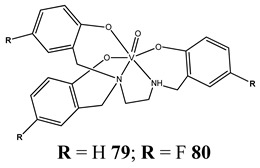	Colon cancer cells HT-29 Ovarian cancer cells OVCAR-3, A2780, A2780cis and A2780adr S cell cycle arrest Induction of apoptosis Cytotoxicity (72 h): HT-29 cells **79: IC_50_ 1.4 ± 0.2 µM** **80: IC_50_ 2.9 ± 0.7 µM** OVCAR-3 cells **79: IC_50_ 0.7 ± 0.3 µM** **80: IC_50_ 0.40 ± 0.04 µM** A2780 cells **79: IC_50_ 0.17 ± 0.07 µM** **80: IC_50_ 0.6 ± 0.1 µM** A2780cis cells **79: IC_50_ 0.29 ± 0.15 µM** **80: IC_50_ 0.8 ± 0.1 µM**	[85]

Bold and Underline: makes Table more readable.

**Table 4 molecules-25-01757-t004:** Structures and mechanism of cell death induced by vanadium compounds (EMT: the epithelial–mesenchymal transition).

Structure	Mechanism	References
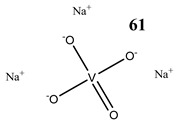	Oral squamous cell carcinoma Cal27 Induction of apoptosis: poly(ADPribose)polymerase cleavage Cytotoxicity (72 h): **IC_50_ 25 μM**	[88]
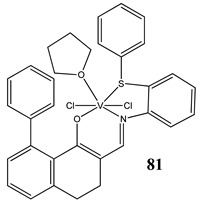	Gastric cancer cells MGC-803 Induction of apoptosis (upregulation of Bax, PARP and caspase-3/9, downregulation of Bcl-2) Prevention from the colony formation, migration and EMT process Cytotoxicity (72 h): **IC_50_ 2.69 μM**	[89]
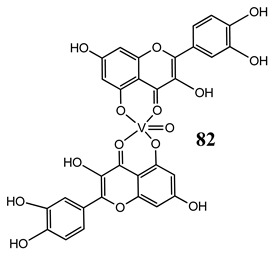	Breast cancer cells MCF-7 Increase in the expression of p53 Decrease in the expression of Akt, mTOR and VEGF Induction of apoptosis (activation of 3 and 9 caspase, DNA fragmentation) In vivo study (Balb/c mice) Increase in apoptotic index Upregulation of Bcl-2 and downregulation of Bax and p53	[90]
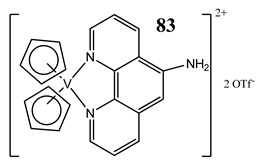	T-leukemic cells (p53 wild-type MOLT-4 and p53-deficient Jurkat) Induction of apoptosis (activation of the caspases 9-intrinsic pathway and 8-extrinsic pathway) Increase in the expression of the tumor-suppressor protein p53 and its form phosphorylated at the serine 15 Cytotoxicity: 24 h: MOLT-4 **IC_50_ 3.1 ± 0.4 μM** Jurkat **IC_50_ 2.9 ± 0.2 μM** 48 h: MOLT-4 **IC_50_ 2.1 ± 0.2 μM** Jurkat **IC_50_ 2.8 ± 0.3 μM** 72 h: MOLT-4 **IC_50_ 2.3 ± 0.2 μM** Jurkat **IC_50_ 1.7 ± 0.1 ± μM**	[91]
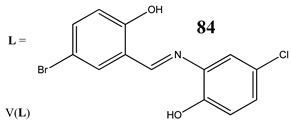	Liver cancer cells HepG2 Induction of apoptosis (using the p53/p21 pathway-dependent way) Decrease in the expression of caspase-8 and Bid	[92]
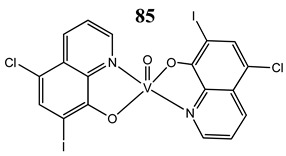	Osteosarcoma cells MG-63 Determination of the relative abundance of 224 proteins Increase in the expression of caspase 3, caspase 6, caspase 7, caspase 10, caspase 11, Bcl-x, DAPK Decrease in the expression of PKB/Akt, DIABLO	[93]
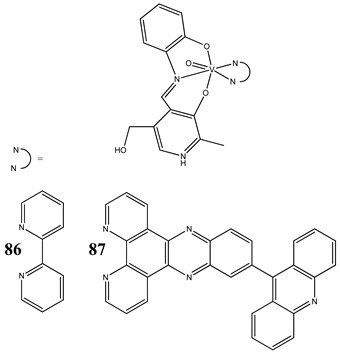	Cervical cancer cells HeLa Breast cancer cells MCF-7 Induction of apoptosis Specific localization to endoplasmic reticulum (ER) Cytotoxicity (48 h): cervical cancer cells HeLa (**86: IC_50_ 44.1 ± 1.8 µM** in visible light **IC_50_ 59.3 ± 1.7 µM in the dark**) (**87: IC_50_ 0.24 ± 0.02 µM** in visible light **IC_50_ > 40 µM** in the dark) breast cancer cells MCF-7 (**86: IC_50_ 53.3 ± 1.9 µM** in visible light **IC_50_ 72.5 ± 2.1 µM** in the dark) (**87: IC_50_ 0.53 ± 0.03 µM** in visible light **IC_50_ > 40 µM** in the dark)	[94]
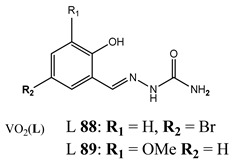	Colon cancer cells HT-29 Induction of necroptosis	[100]
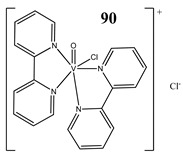	Breast cancer cells MDA-MB-231 G_2_/M cell cycle arrest Induction of autophagy Induction of apoptosis Increase in the expression of Notch 1	[101]
**91** MSAGM:VO MSAGM = Galactomannan from *Schizolobium amazonicum*	Liver cancer cells HepG2 Induction of apoptosis under normoxic conditions (lost under hypoxic conditions) The expressions of anti-apoptotic Mcl-1 and Bcl-XL increased in hypoxia, whereas the expression of pro-apoptotic Bax decreased Induction of autophagy (elimination of the anti-cancer activity with activation of autophagy under conditions of hypoxia)	[102]

Bold and Underline: makes Table more readable.

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
