# Peer review of "Molecular and Cellular Mechanisms of Cytotoxic Activity of Vanadium Compounds against Cancer Cells"

_molecules, 2020, doi:10.3390/molecules25071757_

Round 1

Reviewer 1 Report

Please replace everywhere in the text used term OXIDOVANADIUM and VANADYL. Chemist never use the former, and it will become instantly obvious for the reader that authors are not chemists! Also, include in the Glossary section at the beginning the following: V(4+) is commonly present in chemical compounds as vanadtl cations, VO(2+).

Author Response

“Please replace everywhere in the text used term OXIDOVANADIUM and VANADYL.
Chemist never use the former, and it will become instantly obvious for the
reader that authors are not chemists! Also, include in the Glossary section
at the beginning the following: V(4+) is commonly present in chemical
compounds as vanadtl cations, VO(2+).”

Authors’ response:

We would like to thank the Reviewer for the important comment. We agree with the Reviewer that V(4+) is commonly present in chemical compounds as oxidometal cation, for example VO(2+) or VO2(+).

According to IUPAC Recommendations 2005 (see Nomenclature of Inorganic Chemistry IUPAC RECOMMENDATIONS 2005, RSC Publishing, p. 257 and 335, https://old.iupac.org/publications/books/rbook/Red_Book_2005.pdf)

ending – yl of certain non-systematic names of oxidometal cations, e.g. vanadyl for

oxidovanadium are no longer acceptable.

For this reason we decided to used oxidovanadium instead of vanadyl.

The appropriate changes have been made in the revised version of the manuscript.

Reviewer 2 Report

Authors improved their paper, but it still needs careful editing (see file attached).

Author Response

“Some typing errors:
135 metal complexes(?)  were
158 cleavde
173 Four”
Authors’ response: We would like to thank the Reviewer for these comments. Errors have been corrected.

Table 1, ref. 29: R = CH2C6H5, H ;  NN = bipy (9) or phen (10)
Only two complexes are reported (9 and 10) so eliminate or R = CH2C6H5 or R =
H.

According to the Reviewer remark structure from ref. 29 has been corrected.

“Figure 1  Both ONOO- and O2NOO- are indicated as “peroxynitrite””

Authors’ response: According to nomenclature of reactive species, both of them have the same name (for example: Halliwell, Barry, and John MC Gutteridge. Free radicals in biology and medicine. Oxford University Press, USA, 2015.)

“234 and others (for ex. 399). “In other study” instead of “In another study”
or “In other studies”
237 an
256 activity were
276 Structers
281 Moreover, Incubation”
Authors’ response: We would like to thank the Reviewer for these comments. Errors have been corrected.

Table 2. Structure 45 is repeated in two following lines

Authors’ response: According to the Reviewer remark Table 2 has been corrected.

322 These studies underscores
338 “Some studies … were found…”, instead of “Some studies have found… “

Authors’ response: We would like to thank the Reviewer for these comments. Errors have been corrected.

Table 3 ref. 83. The structure 77 is improbable: N insted of O in the cycle?
According to the Reviewer remark Table 3 has been corrected.

“376 It is better “vanadium complexes” (they are 67 and 68), instead of
“vanadium complex”
382 and 383. The same as above:  “complexes” (86 and 87) and not “complex”
434 desribed”

Authors’ response: We would like to thank the Reviewer for these comments. Errors have been corrected.

“ref. 18. has to be completed (undefined?): FriedrichSchiller-Universitat, M.
A.-; Jena, undefined; 1986, undefined. Spurenelement Symposium: New Trace
Elements/Anke M., Baumann W., Braunlich H., Bruckner C. and Groppel B.
ref. 70. (80-. ) ???
ref. 89. 2019 is in bold, not italics
ref. 109. Therputic”

Authors’ response: We would like to thank the Reviewer for these comments. Errors in literature have been corrected.

Reviewer 3 Report

The manuscript definitely was improved with the incorporation of molecular structures and the summary of cytotoxic damage elicited by each compound. However, there are some issues that still have to be solved before considering the publication of the manuscript.

The authors stated that vanadium organometallic compounds listed in Tables 1 and 2 damage DNA and induce ROS overproduction (page 12, lines 282-285), meanwhile, vanadium organometallic compounds induce apoptosis (page 23, lines 365-385). However, I could not find any vanadium organometallic compound in the whole manuscript. There is no V-C bond in any of the molecular structures depicted or mentioned. The authors must modify this.

The authors mention several times on the manuscript the importance of the chemical form of vanadium-based complexes in determining their mechanism of action, however, considering the existing information on previous reviews that authors list as their references:

i) there is no discussion about the role played by the ligands on the stabilization of vanadium oxidation state and in its modulation of the mechanism of action exerted by the vanadium coordination compounds.
ii) Several V-phenanthroline, V-bipyridine and V-terpyridine derivatives are listed in tables 1, 2 and 3.
a) How these ligands contribute to the redox potential modulation of vanadium compounds?
b) How the presence of those ligands promote redox reactions within the cells and lead to oxidative stress? Are these ligands essential for the reported DNA interaction (Kb) and its subsequent damage?
c) The ligands present in the vanadium coordination sphere promote specific interactions that lead to a different cytotoxic response, especially those with phen, bipy and terpy. It is known that these ligands enhance the interaction with nucleic acids and modulate the redox environment of metal ions.

d) All the above points (a-c) must be associated with the summary of suggested molecular and cellular mechanisms shown in figure 2.

The authors must provide a deep discussion about the aforementioned issues.

Finally, in the conclusions, I assume that the authors associate the molecular and cellular mechanisms of vanadium coordination compounds to the oxidation state of vanadium, coordination sphere, geometrical arrangement of the coordination compound and the cellular typo exposed to them. To support this conclusion, the aforementioned suggested discussion (points i and ii) must be included.

In my opinion, the term "valence of vanadium" is misused, instead, the authors must use the "oxidation state of vanadium".

Minor revisions

Some typo must be corrected

line 276 structure instead structers
line 319 pyridone instead pirydone
and so on

Author Response

“The authors stated that vanadium organometallic compounds listed in Tables 1
and 2 damage DNA and induce ROS overproduction (page 12, lines 282-285),
meanwhile, vanadium organometallic compounds induce apoptosis (page 23, lines
365-385). However, I could not find any vanadium organometallic compound in
the whole manuscript. There is no V-C bond in any of the molecular structures
depicted or mentioned. The authors must modify this.”

Authors’ response: We would like to apologize for the misreading. The term “vanadium organometallic compound” has been changed into “vanadium-based complexes”.

“The authors mention several times on the manuscript the importance of the
chemical form of vanadium-based complexes in determining their mechanism of
action, however, considering the existing information on previous reviews
that authors list as their references:

i) there is no discussion about the role played by the ligands on the
stabilization of vanadium oxidation state and in its modulation of the
mechanism of action exerted by the vanadium coordination compounds

ii) Several V-phenanthroline, V-bipyridine and V-terpyridine derivatives are
listed in tables 1, 2 and 3.
a) How these ligands contribute to the redox potential modulation of vanadium
compounds

b) How the presence of those ligands promote redox reactions within the cells
and lead to oxidative stress? Are these ligands essential for the reported
DNA interaction (Kb) and its subsequent damage

c) The ligands present in the vanadium coordination sphere promote specific
interactions that lead to a different cytotoxic response, especially those
with phen, bipy and terpy. It is known that these ligands enhance the
interaction with nucleic acids and modulate the redox environment of metal

d) All the above points (a-c) must be associated with the summary of
suggested molecular and cellular mechanisms shown in figure 2.
The authors must provide a deep discussion about the aforementioned issues.

Finally, in the conclusions, I assume that the authors associate the
molecular and cellular mechanisms of vanadium coordination compounds to the
oxidation state of vanadium, coordination sphere, geometrical arrangement of
the coordination compound and the cellular typo exposed to them. To support
this conclusion, the aforementioned suggested discussion (points i and ii)
must be included.”

Authors’ response: According to the Reviewer suggestion conclusion section has been rewritten:

The studies described in this review suggest that molecular and cellular mechanisms of vanadium compounds depend on many factors, including the oxidation state of vanadium cation, organic ligands, spatial structure and also type of cancer cell lines. That is why we can observe so many, sometimes mutually exclusive, mechanisms of cytotoxicity.

The literature survey revealed that the chemical form of vanadium-based complexes (oxidation state of vanadium, the type of the ligands and their geometrical arrangement) influences their physicochemical properties and thus their biological properties. For instance, the presence of a strong binding ligand in the coordination sphere of the VO2+ ion hinders the oxidation of the metal ion, V(IV) to V(V) [110]. Furthermore, a nuclease activity of the V-phenanthroline, V-bipyridine and V-terpyridine compounds depends on the number of intercalating heterocyclic moieties. It suggests that the incorporation into coordination sphere of the vanadium cation the appropriate type of ligands may promote redox reactions or enhance the interaction with nucleic acids. This leads to the oxidative stress, DNA damage, cell cycle arrest and ultimately to the cell death (Figure 2).

Although, there are some evidence that the structure and physicochemical properties of the vanadium complexes have the impact on their biological activity, the correlation of chemical form of vanadium-based complexes versus mechanism of action still remains to be elucidated. Moreover, the differences in physicochemical and biological properties of the compounds may stem from very different experimental (chemical and biological) conditions. The results of chemical studies on physicochemical properties of the compounds should be assessed very carefully as the experimental conditions leading to these results are generally very different from biological conditions. Furthermore, there are still very few in vivo studies and the almost complete lack of innovative approach based on targeted therapies. In view of this, further biological research, focusing on a more in-depth analysis of cytotoxic activity and using the most modern techniques, are required.

In conclusion, the above considerations underline the anticancer potential of vanadium-based compounds. Through modification of the chemical form of vanadium-based complexes, we can influence on affinity for DNA, oxidative stress or type of cell death induce by vanadium-based compounds in cancer cells. On the other hand, due to many factors, it is difficult to precisely define structure-activity relations.  

“In my opinion, the term "valence of vanadium" is misused, instead, the
authors must use the "oxidation state of vanadium".”

Authors’ response: We would like to thank the Reviewer for this valuable remark. The term "valence of vanadium" has been corrected in the revised version of the manuscript.

Round 2

Reviewer 3 Report

The authors made all the suggested changes. The manuscript can be published in its actual form.